# FIGHTER: UNVEILING THE GRAPH CONVOLUTIONAL NATURE OF TRANSFORMERS IN TIME SERIES MODELING

## ABSTRACT

Transformers have achieved remarkable success in time series modeling, yet their internal mechanisms remain opaque. This work demystifies the Transformer encoder by establishing its fundamental equivalence to a Graph Convolutional Network (GCN). We show that in the forward pass, the attention distribution matrix serves as a dynamic adjacency matrix, and its composition with subsequent transformations performs computations analogous to graph convolution. Moreover, we demonstrate that in the backward pass, the update dynamics of value and feed-forward projections mirror those of GCN parameters. Building on this unified theoretical reinterpretation, we propose **Fighter** (Flexible Graph Convolutional Transformer), a streamlined architecture that removes redundant linear projections and incorporates multi-hop graph aggregation. This perspective yields an explicit and interpretable representation of temporal dependencies across different scales, naturally expressed as graph edges. Experiments on standard forecasting benchmarks confirm that Fighter achieves competitive performance while providing clearer mechanistic interpretability of its predictions.

## 1 INTRODUCTION AND RELATED WORK

Time series data is prevalent in domains like finance (Tsay, 2005), transportation (Vlahogianni & Karlaftis, 2013), and energy (Wang et al., 2007). Transformers (Vaswani et al., 2017) have emerged as promising solutions for time series modeling due to their ability to capture long-range dependencies (Wen et al., 2023). However, their internal workings remain poorly understood (Rudin, 2019; Zeng et al., 2023; Zhang et al., 2026), raising questions: (1) How does the Transformer process input time series data, particularly the roles of attention and feed-forward networks (FFNs)? (2) What are the update dynamics of its parameters?

To address these, we establish a fundamental equivalence between Transformer encoders and Graph Convolutional Networks (GCNs) (Kipf & Welling, 2017). GCNs model graphs defined by a static adjacency matrix $A_{n \times n}$ and feature matrix $X_{n \times d}$ (Hamilton et al., 2017; Chami et al., 2022), aggregating features across hops (Wu et al., 2019; Zhang et al., 2025). We show Transformers perform analogous operations with a dynamic adjacency.

Related efforts in Transformer-based time series include efficiency improvements like Informer (Zhou et al., 2021), Pyraformer (Liu et al., 2022), Triformer (Cirstea et al., 2022), and PatchTST (Nie et al., 2023), and periodicity modeling like Autoformer (Wu et al., 2021) and FEDformer (Zhou et al., 2022). Recent works like PFformer (Li & Anastasiu, 2025) and PENGUIN (Sun et al., 2025) focus on adaptive settings. However, they lack mechanistic insights.

Graph-based methods like GTS (Shang et al., 2021), MTGNN (Wu et al., 2020), and FourierGNN (Yi et al., 2023) model time series as graphs but often use static adjacencies or limit temporal range. Some treat sequences as graphs (Li et al., 2024; Joshi, 2025), but lack systematic forward/backward analysis or multi-hop integration. Interpretability efforts are post-hoc (Vaswani et al., 2017; Wickstrøm et al., 2020).

Our work bridges Transformers and GCNs with a dynamic temporal adjacency, removes redundancies, and adds multi-hop aggregation for interpretable multi-scale dependencies.

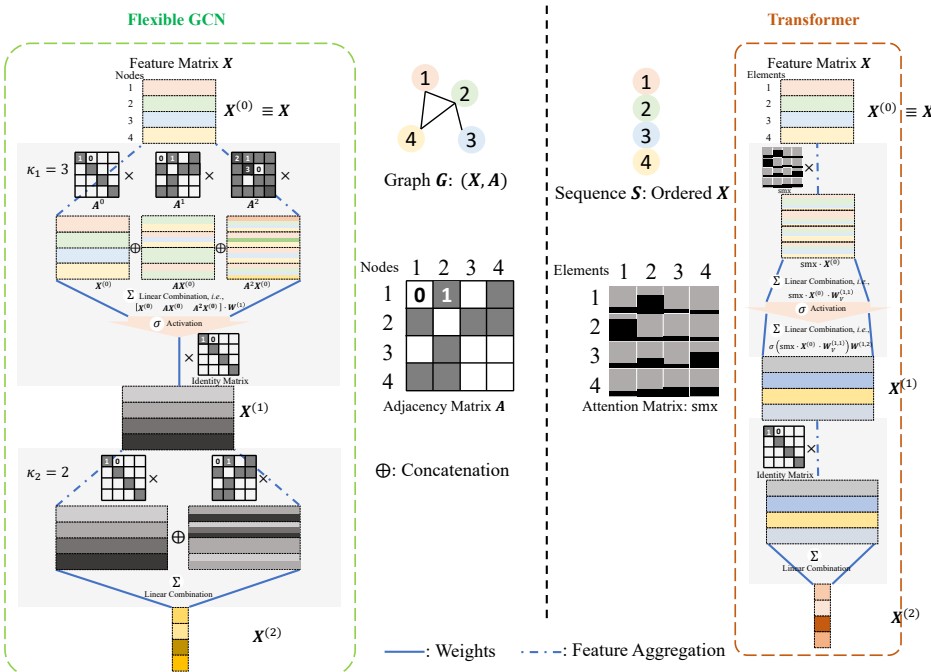

Figure 1: Visual comparison of a Transformer encoder and a flexible GCN, illustrating the equivalence between the Transformer's attention distribution matrix and the GCN's adjacency matrix in performing feature aggregation during the forward pass.

## 2 TRANSFORMERS AS DYNAMIC GRAPH CONVOLUTION FOR TIME SERIES

**Notation**. A graph $G = (X_{n \times d}, A_{n \times n}) \in \mathcal{G}$, where $X_{n \times d}$ has node features $x_i \in \mathbb{R}^d$, $i \in \mathbb{N}_n :=$ $\{1, \ldots, n\}$. A sequence $S_{1:S} = (x_1, \ldots, x_S) \in \mathcal{S}$, with $X_{S \times d}$. A block diagonal matrix with entry matrices $M_1, \ldots, M_m$ is denoted $\mathrm{blkdiag}(M_1, \ldots, M_m)$, and if all $m$ entries are identical, it is simplified to $\mathrm{blkdiag}(M; m)$.

A flexible GCN (Krishnagopal & Ruiz, 2023; Zhang et al., 2025) with multi-hop aggregation is:

$$X_{\mathrm{GCN}}^{(\ell)} = \sigma\left(A^{[\kappa_\ell]}\mathrm{blkdiag}(X_{\mathrm{GCN}}^{(\ell-1)}; \kappa_\ell) \cdot W^{(\ell)}\right), \ell \in \mathbb{N}_{L-1}$$

$$X_{\mathrm{GCN}}^{(L)} = A^{[\kappa_L]}\mathrm{blkdiag}(X_{\mathrm{GCN}}^{(L-1)}; \kappa_L) \cdot W^{(L)}, \tag{1}$$

where $A^{[\kappa]} = [I, A, \ldots, A^{\kappa-1}]$, $W^{(\ell)}$ is $h_{\ell-1} \times h_\ell$, $\kappa_\ell$ is hop distance, $\sigma$ is activation (e.g., ReLU). Single-hop simplifies to:

$$X_{\mathrm{GCN}}^{(\ell)} = \sigma\left(AX_{\mathrm{GCN}}^{(\ell-1)}W^{(\ell)}\right), \ell \in \mathbb{N}_{L-1}. \tag{2}$$

A Transformer encoder (single-head, omitting tokenization, positional encodings, normalization, residuals for clarity) is:

$$X_{\mathrm{Att}}^{(\ell)} = \mathrm{smx}\left(d^{-1/2}X_{\mathrm{FFN}}^{(\ell-1)}W_Q^{(\ell)}\left(X_{\mathrm{FFN}}^{(\ell-1)}W_K^{(\ell)}\right)^\top\right)X_{\mathrm{FFN}}^{(\ell-1)}W_V^{(\ell)}, \ell \in \mathbb{N}_{L-1}$$

$$X_{\mathrm{FFN}}^{(\ell)} = \sigma\left(X_{\mathrm{Att}}^{(\ell)}W_{\mathrm{FFN}}^{(\ell,1)} + b_{\mathrm{FFN}}^{(\ell,1)}\right)W_{\mathrm{FFN}}^{(\ell,2)} + b_{\mathrm{FFN}}^{(\ell,2)}$$

$$X_{\mathrm{TF}}^{(L)} = X_{\mathrm{FFN}}^{(L-1)}W^{(L)}, \tag{3}$$

where smx is row-wise softmax, $W_Q^{(\ell)}, W_K^{(\ell)}, W_V^{(\ell)}$ are query/key/value weights.

### 2.1 FORWARD PASS: THE ATTENTION DISTRIBUTION MATRIX AS A DYNAMIC ADJACENCY

Reformulating the Transformer (omitting biases, defining $W_V^{(\ell,1)} = W_V^{(\ell)}W_{\mathrm{FFN}}^{(\ell,1)}$):

$$X_{\mathrm{TF}}^{(\ell)} = \sigma\left(\mathrm{smx}\left(d^{-1/2}X_{\mathrm{TF}}^{(\ell-1)}W_Q^{(\ell)}\left(X_{\mathrm{TF}}^{(\ell-1)}W_K^{(\ell)}\right)^\top\right)X_{\mathrm{TF}}^{(\ell-1)}W_V^{(\ell,1)}\right) \cdot W_{\mathrm{FFN}}^{(\ell,2)}. \tag{4}$$

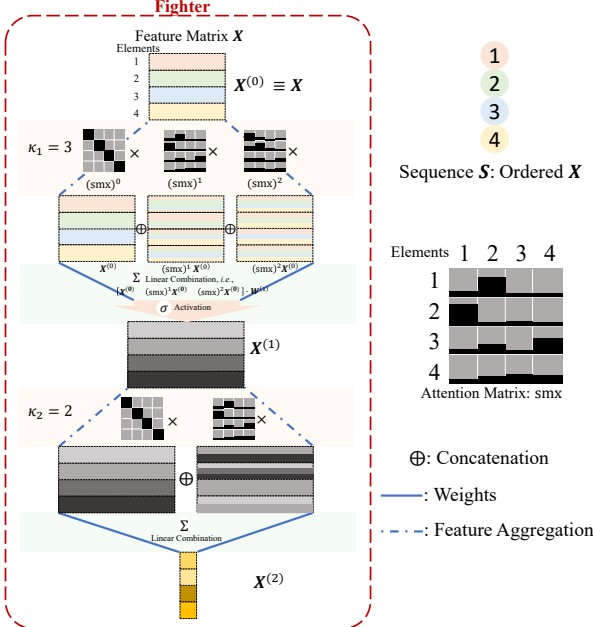

Figure 2: Visual illustration of the Fighter architecture, showcasing its streamlined design with multi-hop graph aggregation and dynamic adjacency matrix for efficient time series modeling.

The softmax term acts as dynamic $A$; $W_{\text{FFN}}^{(\ell,2)}$ is redundant, subsumed by next-layer projections. Comparing to Eq. 2, Transformer is single-hop GCN with learnable adjacency. The derivation of this reformulation is provided in Appendix A.2. A visual comparison of the Transformer encoder and flexible GCN is provided in Figure 1.

**Remark 1.** *This equivalence holds across attention mechanisms, unifying sequence and graph processing.*

## 2.2 BACKWARD PASS: GRADIENT-BASED LEARNING OF DYNAMIC STRUCTURE

For two-layer scalar Transformer:

$$X_{\text{TF}}^{(2)} = \sigma\left(\text{smx}\left(d^{-1/2} X^{(0)} W_Q^{(1)} \left(X^{(0)} W_K^{(1)}\right)^{\top}\right) X^{(0)} W_V^{(1,1)}\right) \cdot W^{(2)}. \tag{5}$$

Gradients for feature parameters:

$$\frac{\partial X_{\text{TF}}^{(2)}}{\partial W^{(2)}} = \sigma\left(\text{smx}\left(d^{-1/2} X^{(0)} W_Q^{(1)} \left(X^{(0)} W_K^{(1)}\right)^{\top}\right) X^{(0)} W_V^{(1,1)}\right),$$

$$\frac{\partial X_{\text{TF}}^{(2)}}{\partial W_{V(:,i)}^{(1,1)}} = \dot{\sigma} \cdot \text{smx}\left(d^{-1/2} X^{(0)} W_Q^{(1)} \left(X^{(0)} W_K^{(1)}\right)^{\top}\right) X^{(0)} \cdot W_{(i,:)}^{(2)}, \tag{6}$$

mirroring GCN gradients.

Query/key gradients enable dynamic adjacency learning (see Appendix A.3 and A.4 for details).

## 2.3 FIGHTER: EFFICIENT MULTI-HOP GRAPH CONVOLUTIONAL TRANSFORMER

Fighter removes redundancies and adds multi-hop:

$$X_{\text{FT}}^{(\ell)} = \sigma\left(\text{smx}^{[\kappa_\ell]}\left(d^{-1/2} X_{\text{FT}}^{(\ell-1)} W_Q^{(\ell)} \left(X_{\text{FT}}^{(\ell-1)} W_K^{(\ell)}\right)^{\top}\right) \text{blkdiag}(X_{\text{FT}}^{(\ell-1)}; \kappa_\ell) \cdot W^{(\ell)}\right)$$

$$X_{\text{FT}}^{(L)} = \text{smx}^{[\kappa_L]}\left(d^{-1/2} X_{\text{FT}}^{(L-1)} W_Q^{(L)} \left(X_{\text{FT}}^{(L-1)} W_K^{(L)}\right)^{\top}\right) \text{blkdiag}(X_{\text{FT}}^{(L-1)}; \kappa_L) \cdot W^{(L)} \tag{7}$$

Table 1: Time series forecasting (MSE/MAE) results on benchmark datasets. 'O' denotes the prediction length. Lower MSE/MAE indicates better performance. The best results are highlighted in bold.

| Dataset | O | Informer | | Reformer | | Transformer | | Fighter ($\kappa = 3$) | |
|---|---|---|---|---|---|---|---|---|---|
| | | MSE | MAE | MSE | MAE | MSE | MAE | MSE | MAE |
| Electricity | 96 | $0.349 \pm 0.002$ | $0.419 \pm 0.004$ | $0.411 \pm 0.009$ | $0.471 \pm 0.006$ | $0.355 \pm 0.013$ | $0.424 \pm 0.011$ | $\mathbf{0.296 \pm 0.007}$ | $\mathbf{0.382 \pm 0.011}$ |
| | 192 | $0.349 \pm 0.007$ | $0.405 \pm 0.027$ | $0.372 \pm 0.009$ | $0.444 \pm 0.003$ | $0.345 \pm 0.017$ | $0.423 \pm 0.016$ | $\mathbf{0.306 \pm 0.004}$ | $\mathbf{0.395 \pm 0.009}$ |
| | 336 | $0.335 \pm 0.004$ | $\mathbf{0.419 \pm 0.003}$ | $0.381 \pm 0.018$ | $0.453 \pm 0.010$ | $0.367 \pm 0.004$ | $0.445 \pm 0.000$ | $\mathbf{0.332 \pm 0.010}$ | $0.422 \pm 0.005$ |
| Weather | 96 | $0.332 \pm 0.036$ | $0.419 \pm 0.022$ | $0.432 \pm 0.053$ | $0.483 \pm 0.037$ | $0.561 \pm 0.201$ | $0.535 \pm 0.088$ | $\mathbf{0.255 \pm 0.011}$ | $\mathbf{0.338 \pm 0.014}$ |
| | 192 | $0.647 \pm 0.174$ | $0.610 \pm 0.097$ | $0.713 \pm 0.059$ | $0.640 \pm 0.037$ | $0.978 \pm 0.297$ | $0.749 \pm 0.121$ | $\mathbf{0.305 \pm 0.003}$ | $\mathbf{0.370 \pm 0.006}$ |
| | 336 | $0.963 \pm 0.031$ | $0.748 \pm 0.001$ | $1.115 \pm 0.169$ | $0.779 \pm 0.090$ | $1.039 \pm 0.589$ | $0.746 \pm 0.228$ | $\mathbf{0.351 \pm 0.005}$ | $\mathbf{0.395 \pm 0.012}$ |
| ETTh1 | 720 | $1.038 \pm 0.086$ | $0.812 \pm 0.020$ | $0.991 \pm 0.003$ | $0.786 \pm 0.014$ | $0.949 \pm 0.024$ | $0.782 \pm 0.007$ | $\mathbf{0.930 \pm 0.049}$ | $\mathbf{0.780 \pm 0.026}$ |

where the value and feed-forward projection is streamlined, and multi-hop aggregation is enabled through the raised attention distribution matrix $\mathrm{smx}^{[\kappa_\ell]}$ and the block-diagonal replication $\mathrm{blkdiag}(\cdot; \kappa_\ell)$. A visualization of Fighter is provided in Figure 2. Fighter is modular, integrable with multi-head, residuals, normalization (Pseudocode 1).

## 3 EMPIRICAL EVALUATION

We evaluate Fighter (with multi-hop parameter $\kappa = 3$) on three standard long-term multivariate forecasting benchmarks: Electricity[1], Weather[2], and ETTh1 (Zhou et al., 2021). Experiments follow established protocols with an input sequence length of 96 and MSE training loss; performance is measured by MSE and MAE.

Table 1 compares Fighter ($\kappa = 3$) against vanilla Transformer (Vaswani et al., 2017), Informer (Zhou et al., 2021), and Reformer (Kitaev et al., 2020). Fighter achieves the best MSE and MAE in the large majority of cases. On Electricity, Fighter obtains the lowest MSE at all horizons and the lowest MAE at 96 and 192 steps: for example, at 96 steps it reaches 0.296 (MSE) / 0.382 (MAE) compared to the next best around 0.349/0.419 (Informer); at 192 steps 0.306/0.395 versus 0.345/0.423 (Transformer); at 336 steps 0.332/0.422, slightly outperforming Informer's 0.335/0.419 in MSE while very close in MAE. On the more challenging Weather dataset, Fighter delivers clearly superior results, especially at longer horizons: 0.255/0.338 (96 steps), 0.305/0.370 (192 steps), and 0.351/0.395 (336 steps), far below the next best values which rise sharply to 0.647–0.963 (MSE) and 0.610–0.748 (MAE) for Informer. On ETTh1 at the 720-step horizon, Fighter records 0.930/0.780, slightly better than the vanilla Transformer's 0.949/0.782 and ahead of Informer (1.038/0.812) and Reformer (0.991/0.786)

These consistent improvements stem from Fighter's streamlined architecture, which eliminates redundant linear projections and the conventional feed-forward networks, while introducing explicit multi-hop graph aggregation. This design more effectively captures long-range and multi-scale temporal dependencies compared to standard attention mechanisms. Compared to vanilla Transformers, Fighter also offers clearer mechanistic interpretability: its dynamic adjacency matrix and explicit $\kappa$-hop aggregation directly expose the multi-scale temporal dependencies in the learned graph, providing more transparent insight into how past time steps contribute to predictions.

## 4 CONCLUDING REMARKS AND FUTURE WORK

This work establishes a fundamental equivalence between Transformer encoders and GCNs in time series, demonstrating that the self-attention mechanism naturally induces a dynamic temporal adjacency matrix while subsequent transformations perform graph-like feature aggregation in both forward and backward passes. Based on this view, we propose Fighter, a streamlined model that removes redundant projections while adding explicit multi-hop graph aggregation. Empirical validation shows that Fighter achieves highly competitive performance, with substantial gains on challenging datasets and longer horizons. Crucially, Fighter delivers clearer mechanistic interpretability than vanilla Transformers by explicitly revealing multi-scale temporal dependencies through its dynamic adjacency and controllable $\kappa$-hop aggregation, providing a more transparent and principled understanding of attention-based sequence modeling.

---

[1] https://archive.ics.uci.edu/ml/datasets/ElectricityLoadDiagrams20112014
[2] https://www.bgc-jena.mpg.de/wetter

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

# Appendix

## A ADDITIONAL DISCUSSIONS

### A.1 NOTATION OVERVIEW

| Notation | Description |
|---|---|
| $\boldsymbol{G} \in \mathcal{G}$ | Graph with $n$ nodes, node features $\boldsymbol{X}_{n \times d}$, and adjacency matrix $\boldsymbol{A}_{n \times n}$ |
| $\boldsymbol{X}_{n \times d}$ | $n \times d$ node feature matrix; rows are node features $\boldsymbol{x}_i \in \mathbb{R}^d$, $i \in \mathbb{N}_n$ |
| $\boldsymbol{A}_{n \times n}$ | $n \times n$ adjacency matrix encoding edges |
| $\mathbb{N}_n \coloneqq \{1, \ldots, n\}$ | Set of node indices |
| $\mathbb{N}_S \coloneqq \{1, \ldots, S\}$ | Set of sequence indices |
| $\boldsymbol{S}_{1:S} \in \mathcal{S}$ | Sequence of length $S$ with features $\boldsymbol{x}_s \in \mathbb{R}^d$, $s \in \mathbb{N}_S$ |
| $\boldsymbol{X}_{S \times d}$ | $S \times d$ sequence feature matrix |
| $\boldsymbol{x} = [x_1, \ldots, x_d]$ | Feature vector as row vector (transpose denotes column form) |
| $\boldsymbol{X}_{(i,:)}$ | $i$-th row of feature matrix $\boldsymbol{X}$ (i.e., $\boldsymbol{e}_i^\top \boldsymbol{X}$) |
| $\boldsymbol{X}_{(:,j)}$ | $j$-th column of feature matrix $\boldsymbol{X}$ (i.e., $\boldsymbol{X} \boldsymbol{e}_j$) |
| $\boldsymbol{e}_i$ | Standard basis vector with 1 in $i$-th position, 0 elsewhere |
| $\boldsymbol{I}$ | Identity matrix |
| $\mathrm{blkdiag}(a_1, \ldots, a_m)$ | Diagonal matrix with entries $a_1, \ldots, a_m$ |
| $\mathrm{blkdiag}(a; m)$ | Diagonal matrix with $m$ identical entries $a$ on the diagonal |

Table 2: Summary of Key Notations.

### A.2 REFORMULATION OF A TRANSFORMER ENCODER

Omitting bias terms, Equation 3 for a Transformer encoder can be expressed as

$$\boldsymbol{X}_{\mathrm{Att}}^{(\ell)} = \mathrm{smx}\left( d^{-1/2} \boldsymbol{X}_{\mathrm{FFN}}^{(\ell-1)} \boldsymbol{W}_Q^{(\ell)} \left( \boldsymbol{X}_{\mathrm{FFN}}^{(\ell-1)} \boldsymbol{W}_K^{(\ell)} \right)^\top \right) \boldsymbol{X}_{\mathrm{FFN}}^{(\ell-1)} \boldsymbol{W}_V^{(\ell)} \tag{8}$$

$$\boldsymbol{X}_{\mathrm{FFN}}^{(\ell)} = \sigma\left( \boldsymbol{X}_{\mathrm{Att}}^{(\ell)} \boldsymbol{W}_{\mathrm{FFN}}^{(\ell,1)} \right) \boldsymbol{W}_{\mathrm{FFN}}^{(\ell,2)} \tag{9}$$

$$\boldsymbol{X}_{\mathrm{Att}}^{(L)} = \boldsymbol{X}_{\mathrm{FFN}}^{(L-1)} \boldsymbol{W}^{(L)}, \tag{10}$$

Substituting Equation 8 into 9 yields:

$$\boldsymbol{X}_{\mathrm{FFN}}^{(\ell)} = \sigma\left( \mathrm{smx}\left( d^{-1/2} \boldsymbol{X}_{\mathrm{FFN}}^{(\ell-1)} \boldsymbol{W}_Q^{(\ell)} \left( \boldsymbol{X}_{\mathrm{FFN}}^{(\ell-1)} \boldsymbol{W}_K^{(\ell)} \right)^\top \right) \boldsymbol{X}_{\mathrm{FFN}}^{(\ell-1)} \boldsymbol{W}_V^{(\ell)} \boldsymbol{W}_{\mathrm{FFN}}^{(\ell,1)} \right) \boldsymbol{W}_{\mathrm{FFN}}^{(\ell,2)} \tag{11}$$

Dropping the subscript for the hidden feature $\boldsymbol{X}_{\mathrm{FFN}}^{(\ell-1)}$ (denoted as $\boldsymbol{X}^{(\ell-1)}$ for brevity) and defining the composite matrix $\boldsymbol{W}_V^{(\ell,1)} \coloneqq \boldsymbol{W}_V^{(\ell)} \boldsymbol{W}_{\mathrm{FFN}}^{(\ell,1)}$, this simplifies to:

$$\boldsymbol{X}^{(\ell)} = \sigma\left( \mathrm{smx}\left( d^{-1/2} \boldsymbol{X}^{(\ell-1)} \boldsymbol{W}_Q^{(\ell)} \left( \boldsymbol{X}^{(\ell-1)} \boldsymbol{W}_K^{(\ell)} \right)^\top \right) \boldsymbol{X}^{(\ell-1)} \boldsymbol{W}_V^{(\ell,1)} \right) \boldsymbol{W}_{\mathrm{FFN}}^{(\ell,2)}. \tag{12}$$

Thus, the Transformer encoder, derived from Equation 3, is:

$$\boldsymbol{X}^{(\ell)} = \sigma\left( \mathrm{smx}\left( d^{-1/2} \boldsymbol{X}^{(\ell-1)} \boldsymbol{W}_Q^{(\ell)} \left( \boldsymbol{X}^{(\ell-1)} \boldsymbol{W}_K^{(\ell)} \right)^\top \right) \boldsymbol{X}^{(\ell-1)} \boldsymbol{W}_V^{(\ell,1)} \right) \boldsymbol{W}_{\mathrm{FFN}}^{(\ell,2)}$$

$$\boldsymbol{X}^{(L)} = \boldsymbol{X}^{(L-1)} \boldsymbol{W}^{(L)}, \tag{13}$$

### A.3 DERIVATION OF GRADIENTS FOR GRAPH CONVOLUTIONAL NETWORK

For a graph convolutional network (GCN), as defined by Equation 1, a two-layer architecture with multi-hop aggregation can be expressed as:

$$\boldsymbol{X}_{\mathrm{GCN}}^{(2)} = \boldsymbol{A}^{[\kappa_2]} \mathrm{blkdiag}\left( \sigma\left( \boldsymbol{A}^{[\kappa_1]} \mathrm{blkdiag}(\boldsymbol{X}^{(0)}; \kappa_1) \cdot \boldsymbol{W}^{(1)} \right); \kappa_2 \right) \cdot \boldsymbol{W}^{(2)}. \tag{14}$$

Using the chain rule, we compute the derivative of $\boldsymbol{X}_{\text{GCN}}^{(2)}$ with respect to the second-layer weights $\boldsymbol{W}^{(2)}$, a vector of dimension $\kappa_2 h_1$, as follows:

$$
\begin{aligned}
\frac{\partial \boldsymbol{X}_{\text{GCN}}^{(2)}}{\partial \boldsymbol{W}^{(2)}} &= \frac{\partial \boldsymbol{A}^{[\kappa_2]}\text{blkdiag}(\boldsymbol{X}^{(1)};\kappa_2)\cdot \boldsymbol{W}^{(2)}}{\partial \boldsymbol{W}^{(2)}} \\
&= \boldsymbol{A}^{[\kappa_2]}\text{blkdiag}(\boldsymbol{X}^{(1)};\kappa_2) \\
&= \underbrace{\overbrace{\boldsymbol{A}^{[\kappa_2]}}^{\text{size: } n\times\kappa_2 n}\ \overbrace{\text{blkdiag}(\sigma(\boldsymbol{A}^{[\kappa_1]}\text{blkdiag}(\boldsymbol{X}^{(0)};\kappa_1)\boldsymbol{W}^{(1)});\kappa_2))}^{\text{size: }\kappa_2 n\times\kappa_2 h_1}}_{\text{size: }1\times\kappa_2 h_1}.
\end{aligned}
\tag{15}
$$

The derivative with respect to the first-layer weights is more involved. For $i \in \mathbb{N}_{h_1}$

$$
\begin{aligned}
\frac{\partial \boldsymbol{X}_{\text{GCN}}^{(2)}}{\partial \boldsymbol{W}_{(:,i)}^{(1)}} &= \frac{\partial \boldsymbol{A}^{[\kappa_2]}\text{blkdiag}(\boldsymbol{X}^{(1)}\boldsymbol{e}_i\boldsymbol{e}_i^\top;\kappa_2)\cdot \boldsymbol{W}^{(2)}}{\partial \boldsymbol{W}_{(:,i)}^{(1)}} \\
&= \frac{\partial \boldsymbol{A}^{[\kappa_2]}\overbrace{\text{blkdiag}(\boldsymbol{X}^{(1)}\boldsymbol{e}_i;\kappa_2)}^{\text{size: }\kappa_2 n\times\kappa_2}\cdot\overbrace{\text{blkdiag}(\boldsymbol{e}_i^\top;\kappa_2)}^{\text{size: }\kappa_2\times\kappa_2 h_1}\boldsymbol{W}^{(2)}}{\partial \boldsymbol{W}_{(:,i)}^{(1)}} \\
&= \frac{\partial \boldsymbol{A}^{[\kappa_2]}\overbrace{\text{blkdiag}(\boldsymbol{X}^{(1)}\boldsymbol{e}_i;\kappa_2)}^{\text{size: }\kappa_2 n\times\kappa_2}\cdot\overbrace{\begin{pmatrix}\boldsymbol{W}_{(i-h_1+h_1)}^{(2)}\\ \cdots\\ \boldsymbol{W}_{(i-h_1+\kappa_2 h_1)}^{(2)}\end{pmatrix}}^{\text{size: }\kappa_2\times 1}}{\partial \boldsymbol{W}_{(:,i)}^{(1)}} \\
&= \frac{\partial \boldsymbol{A}^{[\kappa_2]}\overbrace{\begin{pmatrix}\boldsymbol{X}^{(1)}\boldsymbol{e}_i\boldsymbol{W}_{(i-h_1+h_1)}^{(2)}\\ \cdots\\ \boldsymbol{X}^{(1)}\boldsymbol{e}_i\boldsymbol{W}_{(i-h_1+\kappa_2 h_1)}^{(2)}\end{pmatrix}}^{\text{size: }\kappa_2 n\times 1}}{\partial \boldsymbol{W}_{(:,i)}^{(1)}} \\
&= \boldsymbol{A}^{[\kappa_2]}\begin{pmatrix}\frac{\partial \boldsymbol{X}^{(1)}\boldsymbol{e}_i}{\partial \boldsymbol{W}_{(:,i)}^{(1)}}\boldsymbol{W}_{(i-h_1+h_1)}^{(2)}\\ \cdots\\ \frac{\partial \boldsymbol{X}^{(1)}\boldsymbol{e}_i}{\partial \boldsymbol{W}_{(:,i)}^{(1)}}\boldsymbol{W}_{(i-h_1+\kappa_2 h_1)}^{(2)}\end{pmatrix} \\
&= \underbrace{\overbrace{\boldsymbol{A}^{[\kappa_2]}}^{\text{size: }n\times\kappa_2 n}\overbrace{\begin{pmatrix}\dot\sigma\cdot \boldsymbol{A}^{[\kappa_1]}\text{blkdiag}(\boldsymbol{X}^{(0)};\kappa_1)\cdot \boldsymbol{W}_{(i-h_1+h_1)}^{(2)}\\ \cdots\\ \dot\sigma\cdot \boldsymbol{A}^{[\kappa_1]}\text{blkdiag}(\boldsymbol{X}^{(0)};\kappa_1)\cdot \boldsymbol{W}_{(i-h_1+\kappa_2 h_1)}^{(2)}\end{pmatrix}}^{\text{size: }\kappa_2 n\times\kappa_1 h_0}}_{\text{size: }1\times\kappa_1 h_0},
\end{aligned}
\tag{16}
$$

where the derivative of the activation function is defined as $\dot\sigma = \frac{\partial\sigma(\boldsymbol{A}^{[\kappa_1]}\text{blkdiag}(\boldsymbol{X}^{(0)};\kappa_1)\boldsymbol{W}_{(:,i)}^{(1)})}{\partial \boldsymbol{A}^{[\kappa_1]}\text{blkdiag}(\boldsymbol{X}^{(0)};\kappa_1)\boldsymbol{W}_{(:,i)}^{(1)}}$.
When using the ReLU activation function, this simplifies to $\dot\sigma \cdot \boldsymbol{A}^{[\kappa_1]}\text{blkdiag}(\boldsymbol{X}^{(0)};\kappa_1) = \sigma\left(\boldsymbol{A}^{[\kappa_1]}\text{blkdiag}(\boldsymbol{X}^{(0)};\kappa_1)\right)$.

Thus, the gradients with respect to $\boldsymbol{W}^{(2)}$ and $\boldsymbol{W}^{(1)}_{(:,i)}$ (for $i \in \mathbb{N}_{h_1}$) are:

$$
\frac{\partial \boldsymbol{X}^{(2)}_{\text{GCN}}}{\partial \boldsymbol{W}^{(2)}} = \underbrace{\overbrace{\boldsymbol{A}^{[\kappa_2]}}^{n \times \kappa_2 n} \overbrace{\text{blkdiag}(\sigma(\boldsymbol{A}^{[\kappa_1]}\text{blkdiag}(\boldsymbol{X}^{(0)}; \kappa_1)\boldsymbol{W}^{(1)}); \kappa_2)}^{\kappa_2 n \times \kappa_2 h_1}}_{1 \times \kappa_2 h_1},
$$

$$
\frac{\partial \boldsymbol{X}^{(2)}_{\text{GCN}}}{\partial \boldsymbol{W}^{(1)}_{(:,i)}} = \underbrace{\overbrace{\boldsymbol{A}^{[\kappa_2]}}^{n \times \kappa_2 n} \overbrace{\begin{pmatrix} \dot{\sigma} \cdot \boldsymbol{A}^{[\kappa_1]}\text{blkdiag}(\boldsymbol{X}^{(0)}; \kappa_1) \cdot \boldsymbol{W}^{(2)}_{(i-h_1+h_1)} \\ \cdots \\ \dot{\sigma} \cdot \boldsymbol{A}^{[\kappa_1]}\text{blkdiag}(\boldsymbol{X}^{(0)}; \kappa_1) \cdot \boldsymbol{W}^{(2)}_{(i-h_1+\kappa_2 h_1)} \end{pmatrix}}^{\kappa_2 n \times \kappa_1 h_0}}_{1 \times \kappa_1 h_0}. \tag{17}
$$

When restricted to single-hop aggregation (*i.e.*, using only the $\boldsymbol{A}^1$ term), the expressions simplify to:

$$
\frac{\partial \boldsymbol{X}^{(2)}_{\text{GCN}}}{\partial \boldsymbol{W}^{(2)}} = \sigma(\boldsymbol{A}\boldsymbol{X}^{(0)}\boldsymbol{W}^{(1)}), \qquad \frac{\partial \boldsymbol{X}^{(2)}_{\text{GCN}}}{\partial \boldsymbol{W}^{(1)}_{(:,i)}} = \dot{\sigma}\boldsymbol{A}\boldsymbol{X}^{(0)}\boldsymbol{W}^{(2)}_{(i,:)}. \tag{18}
$$

Here, $\boldsymbol{A}$ is a predefined adjacency matrix encoding the graph structure, and $\boldsymbol{W}^{(\ell)}$ governs feature transformation.

### A.4 DERIVATION OF GRADIENTS FOR TRANSFORMER ENCODER

For the Transformer encoder, we simplify Equation 4 by removing the unnecessary linear projection $\boldsymbol{W}^{(\ell,2)}_{\text{FFN}}$ and fixing $L = 2$, resulting in the final layer output:

$$
\boldsymbol{X}^{(2)}_{\text{TF}} = \sigma\left(\text{smx}\left(d^{-1/2}\boldsymbol{X}^{(0)}\boldsymbol{W}^{(1)}_Q\big(\boldsymbol{X}^{(0)}\boldsymbol{W}^{(1)}_K\big)^\top\right)\boldsymbol{X}^{(0)}\boldsymbol{W}^{(1,1)}_V\right) \cdot \boldsymbol{W}^{(2)}, \tag{19}
$$

#### A.4.1 GRADIENTS W.R.T. FEATURE TRANSFORMATION PARAMETERS $\boldsymbol{W}^{(2)}$ AND $\boldsymbol{W}^{(1,1)}_{V(:,i)}$

The gradient with respect to $\boldsymbol{W}^{(2)}$ is given by:

$$
\frac{\partial \boldsymbol{X}^{(2)}_{\text{TF}}}{\partial \boldsymbol{W}^{(2)}} = \sigma\left(\text{smx}\left(d^{-1/2}\boldsymbol{X}^{(0)}\boldsymbol{W}^{(1)}_Q\big(\boldsymbol{X}^{(0)}\boldsymbol{W}^{(1)}_K\big)^\top\right)\boldsymbol{X}^{(0)}\boldsymbol{W}^{(1,1)}_V\right) \tag{20}
$$

Similarly, the gradient with respect to $\boldsymbol{W}^{(1,1)}_{V(:,i)}$ ($i \in \mathbb{N}_{h_1}$) is

$$
\begin{aligned}
\frac{\partial \boldsymbol{X}^{(2)}_{\text{TF}}}{\partial \boldsymbol{W}^{(1,1)}_{V(:,i)}} &= \frac{\partial \sigma\left(\text{smx}\left(d^{-1/2}\boldsymbol{X}^{(0)}\boldsymbol{W}^{(1)}_Q\big(\boldsymbol{X}^{(0)}\boldsymbol{W}^{(1)}_K\big)^\top\right)\boldsymbol{X}^{(0)}\boldsymbol{W}^{(1,1)}_V\right) \cdot \boldsymbol{W}^{(2)}}{\partial \boldsymbol{W}^{(1,1)}_{V(:,i)}} \\
&= \dot{\sigma} \cdot \frac{\partial \text{smx}\left(d^{-1/2}\boldsymbol{X}^{(0)}\boldsymbol{W}^{(1)}_Q\big(\boldsymbol{X}^{(0)}\boldsymbol{W}^{(1)}_K\big)^\top\right)\boldsymbol{X}^{(0)}\boldsymbol{W}^{(1,1)}_V}{\partial \boldsymbol{W}^{(1,1)}_{V(:,i)}} \cdot \boldsymbol{W}^{(2)} \\
&= \dot{\sigma} \cdot \text{smx}\left(d^{-1/2}\boldsymbol{X}^{(0)}\boldsymbol{W}^{(1)}_Q\big(\boldsymbol{X}^{(0)}\boldsymbol{W}^{(1)}_K\big)^\top\right)\boldsymbol{X}^{(0)}\boldsymbol{e}_i^\top \cdot \boldsymbol{W}^{(2)} \\
&= \dot{\sigma} \cdot \text{smx}\left(d^{-1/2}\boldsymbol{X}^{(0)}\boldsymbol{W}^{(1)}_Q\big(\boldsymbol{X}^{(0)}\boldsymbol{W}^{(1)}_K\big)^\top\right)\boldsymbol{X}^{(0)} \cdot \boldsymbol{W}^{(2)}_{(i,:)}, \tag{21}
\end{aligned}
$$

where $\dot{\sigma}$ denotes the derivative of the activation function.

### A.4.2 Gradients with Respect to Query and Key Weight Matrices

Computing the derivative of $\boldsymbol{X}_{\mathrm{TF}}^{(2)}$ with respect to the **query weight matrix** is more complex. For $i \in \mathbb{N}_p$,

$$
\begin{aligned}
\frac{\partial \boldsymbol{X}_{\mathrm{TF}}^{(2)}}{\partial \boldsymbol{W}_{Q\,(:,i)}^{(1)}}
&= \frac{\partial\, \sigma\Big(\mathrm{smx}\Big(d^{-1/2}\boldsymbol{X}^{(0)}\boldsymbol{W}_Q^{(1)}\big(\boldsymbol{X}^{(0)}\boldsymbol{W}_K^{(1)}\big)^{\top}\Big)\boldsymbol{X}^{(0)}\boldsymbol{W}_V^{(1,1)}\Big)\cdot\boldsymbol{W}^{(2)}}{\partial \boldsymbol{W}_{Q\,(:,i)}^{(1)}} \\[2mm]
&= \dot{\sigma}\,\frac{\partial\, \mathrm{smx}\Big(d^{-1/2}\boldsymbol{X}^{(0)}\boldsymbol{W}_Q^{(1)}\big(\boldsymbol{X}^{(0)}\boldsymbol{W}_K^{(1)}\big)^{\top}\Big)\boldsymbol{X}^{(0)}\boldsymbol{W}_V^{(1,1)}}{\partial \boldsymbol{W}_{Q\,(:,i)}^{(1)}}\cdot\boldsymbol{W}^{(2)} \\[2mm]
&= \dot{\sigma}\,\frac{\partial\, \mathrm{smx}\Big(d^{-1/2}\boldsymbol{X}^{(0)}\boldsymbol{W}_Q^{(1)}\boldsymbol{e}_i\boldsymbol{e}_i^{\top}\big(\boldsymbol{X}^{(0)}\boldsymbol{W}_K^{(1)}\big)^{\top}\Big)\boldsymbol{X}^{(0)}\boldsymbol{W}_V^{(1,1)}}{\partial \boldsymbol{W}_{Q\,(:,i)}^{(1)}}\cdot\boldsymbol{W}^{(2)} \\[2mm]
&= \begin{bmatrix}
\dot{\sigma}_1\,\dfrac{\partial\, \mathrm{smx}\Big(d^{-1/2}\boldsymbol{X}_{(1,:)}^{(0)}\boldsymbol{W}_{Q\,(:,i)}^{(1)}\boldsymbol{W}_{K\,(:,i)}^{(1)}{}^{\top}\boldsymbol{X}^{(0)\top}\Big)\boldsymbol{X}^{(0)}\boldsymbol{W}_V^{(1,1)}}{\partial \boldsymbol{W}_{Q\,(:,i)}^{(1)}}\cdot\boldsymbol{W}^{(2)} \\
\cdots \\
\dot{\sigma}_S\,\dfrac{\partial\, \mathrm{smx}\Big(d^{-1/2}\boldsymbol{X}_{(S,:)}^{(0)}\boldsymbol{W}_{Q\,(:,i)}^{(1)}\boldsymbol{W}_{K\,(:,i)}^{(1)}{}^{\top}\boldsymbol{X}^{(0)\top}\Big)\boldsymbol{X}^{(0)}\boldsymbol{W}_V^{(1,1)}}{\partial \boldsymbol{W}_{Q\,(:,i)}^{(1)}}\cdot\boldsymbol{W}^{(2)}
\end{bmatrix} \\[2mm]
&= \begin{bmatrix}
\dot{\sigma}_1\,\dfrac{\partial\, \mathrm{smx}\Big(d^{-1/2}\boldsymbol{X}_{(1,:)}^{(0)}\boldsymbol{W}_{Q\,(:,i)}^{(1)}\boldsymbol{W}_{K\,(:,i)}^{(1)}{}^{\top}\boldsymbol{X}^{(0)\top}\Big)}{\partial \boldsymbol{W}_{Q\,(:,i)}^{(1)}}\cdot\boldsymbol{X}^{(0)}\boldsymbol{W}_V^{(1,1)}\boldsymbol{W}^{(2)} \\
\cdots \\
\dot{\sigma}_S\,\dfrac{\partial\, \mathrm{smx}\Big(d^{-1/2}\boldsymbol{X}_{(S,:)}^{(0)}\boldsymbol{W}_{Q\,(:,i)}^{(1)}\boldsymbol{W}_{K\,(:,i)}^{(1)}{}^{\top}\boldsymbol{X}^{(0)\top}\Big)}{\partial \boldsymbol{W}_{Q\,(:,i)}^{(1)}}\cdot\boldsymbol{X}^{(0)}\boldsymbol{W}_V^{(1,1)}\boldsymbol{W}^{(2)}
\end{bmatrix}
\end{aligned}
\tag{22}
$$

To unpack this further, we analyze it row by row. For the $j$-th row ($j \in \mathbb{N}_S$) in Equation 22, the key derivative term is $\dfrac{\partial\, \mathrm{smx}\Big(d^{-1/2}\boldsymbol{X}_{(j,:)}^{(0)}\boldsymbol{W}_{Q\,(:,i)}^{(1)}\boldsymbol{W}_{K\,(:,i)}^{(1)}{}^{\top}\boldsymbol{X}^{(0)\top}\Big)}{\partial \boldsymbol{W}_{Q\,(:,i)}^{(1)}}$

$$
\begin{aligned}
&\frac{\partial\, \mathrm{smx}\Big(d^{-1/2}\boldsymbol{X}_{(j,:)}^{(0)}\boldsymbol{W}_{Q\,(:,i)}^{(1)}\boldsymbol{W}_{K\,(:,i)}^{(1)}{}^{\top}\boldsymbol{X}^{(0)\top}\Big)}{\partial \boldsymbol{W}_{Q\,(:,i)}^{(1)}} \\[2mm]
&= \frac{\partial\, d^{-1/2}\boldsymbol{X}_{(j,:)}^{(0)}\boldsymbol{W}_{Q\,(:,i)}^{(1)}\boldsymbol{W}_{K\,(:,i)}^{(1)}{}^{\top}\boldsymbol{X}^{(0)\top}}{\partial \boldsymbol{W}_{Q\,(:,i)}^{(1)}}\frac{\partial\, \mathrm{smx}\Big(d^{-1/2}\boldsymbol{X}_{(j,:)}^{(0)}\boldsymbol{W}_{Q\,(:,i)}^{(1)}\boldsymbol{W}_{K\,(:,i)}^{(1)}{}^{\top}\boldsymbol{X}^{(0)\top}\Big)}{\partial\, d^{-1/2}\boldsymbol{X}_{(j,:)}^{(0)}\boldsymbol{W}_{Q\,(:,i)}^{(1)}\boldsymbol{W}_{K\,(:,i)}^{(1)}{}^{\top}\boldsymbol{X}^{(0)\top}} \\[2mm]
&= \boldsymbol{X}_{(j,:)}^{(0)}\cdot\frac{1}{\sqrt{d}}\boldsymbol{W}_{K\,(:,i)}^{(1)}{}^{\top}\boldsymbol{X}^{(0)\top}\frac{\partial\, \mathrm{smx}\Big(d^{-1/2}\boldsymbol{X}_{(j,:)}^{(0)}\boldsymbol{W}_{Q\,(:,i)}^{(1)}\boldsymbol{W}_{K\,(:,i)}^{(1)}{}^{\top}\boldsymbol{X}^{(0)\top}\Big)}{\partial\, d^{-1/2}\boldsymbol{X}_{(j,:)}^{(0)}\boldsymbol{W}_{Q\,(:,i)}^{(1)}\boldsymbol{W}_{K\,(:,i)}^{(1)}{}^{\top}\boldsymbol{X}^{(0)\top}} \\[2mm]
&= \boldsymbol{X}_{(j,:)}^{(0)}\cdot\frac{1}{\sqrt{d}}\boldsymbol{W}_{K\,(:,i)}^{(1)}{}^{\top}\boldsymbol{X}^{(0)\top}\frac{\partial\left(\dfrac{\exp\Big(d^{-1/2}\boldsymbol{X}_{(j,:)}^{(0)}\boldsymbol{W}_{Q\,(:,i)}^{(1)}\boldsymbol{W}_{K\,(:,i)}^{(1)}{}^{\top}\boldsymbol{X}^{(0)\top}\Big)}{\mathbf{1}^{\top}\exp\Big(d^{-1/2}\boldsymbol{X}_{(j,:)}^{(0)}\boldsymbol{W}_{Q\,(:,i)}^{(1)}\boldsymbol{W}_{K\,(:,i)}^{(1)}{}^{\top}\boldsymbol{X}^{(0)\top}\Big)}\right)}{\partial\, d^{-1/2}\boldsymbol{X}_{(j,:)}^{(0)}\boldsymbol{W}_{Q\,(:,i)}^{(1)}\boldsymbol{W}_{K\,(:,i)}^{(1)}{}^{\top}\boldsymbol{X}^{(0)\top}} \\[2mm]
&= \boldsymbol{X}_{(j,:)}^{(0)}\cdot\Big(\frac{1}{\sqrt{d}}\boldsymbol{W}_{K\,(:,i)}^{(1)}{}^{\top}\boldsymbol{X}^{(0)\top}\mathrm{blkdiag}\big(\mathrm{smx}(\boldsymbol{X}_{(j,:)}^{(0)}\boldsymbol{W}_{Q\,(:,i)}^{(1)}\boldsymbol{W}_{K\,(:,i)}^{(1)}{}^{\top}\boldsymbol{X}^{(0)\top}/\sqrt{d})\big) \\
&\qquad -\frac{1}{\sqrt{d}}\boldsymbol{W}_{K\,(:,i)}^{(1)}{}^{\top}\boldsymbol{X}^{(0)\top}\big(\mathrm{smx}(\boldsymbol{X}_{(j,:)}^{(0)}\boldsymbol{W}_{Q\,(:,i)}^{(1)}\boldsymbol{W}_{K\,(:,i)}^{(1)}{}^{\top}\boldsymbol{X}^{(0)\top}/\sqrt{d})\big)^{\top}\mathrm{smx}(\boldsymbol{X}_{(j,:)}^{(0)}\boldsymbol{W}_{Q\,(:,i)}^{(1)}\boldsymbol{W}_{K\,(:,i)}^{(1)}{}^{\top}\boldsymbol{X}^{(0)\top}/\sqrt{d})\Big)
\end{aligned}
\tag{23}
$$

The right-hand side of Equation 23 simplifies to:

$$
\boldsymbol{X}_{(j,:)}^{(0)} \cdot \Bigg( \underbrace{d^{-1/2}}_{1\times 1} \underbrace{\boldsymbol{W}_{K\,(:,i)}^{(1)}}_{1\times d}{}^{\top} \overbrace{\boldsymbol{X}^{(0)}{}^{\top}}^{d\times S} \mathrm{blkdiag}\Big( \mathrm{smx}(\overbrace{\boldsymbol{X}_{(j,:)}^{(0)}}^{1\times d} \overbrace{\boldsymbol{W}_{Q\,(:,i)}^{(1)}}^{d\times 1} \overbrace{\boldsymbol{W}_{K\,(:,i)}^{(1)}{}^{\top}}^{1\times d} \overbrace{\boldsymbol{X}^{(0)}{}^{\top}}^{d\times S} /\sqrt{d}) \Big)
$$
$$
\underbrace{\phantom{\mathrm{blkdiag}\Big(\mathrm{smx}(\boldsymbol{X}_{(j,:)}^{(0)}\boldsymbol{W}_{Q}^{(1)})\Big)}}_{S\times S}
$$
$$
-\underbrace{d^{-1/2}}_{1\times 1}\underbrace{\boldsymbol{W}_{K\,(:,i)}^{(1)}}_{1\times d}{}^{\top}\overbrace{\boldsymbol{X}^{(0)}{}^{\top}}^{d\times S}\underbrace{\Big(\mathrm{smx}(\overbrace{\boldsymbol{X}_{(j,:)}^{(0)}}^{1\times d}\overbrace{\boldsymbol{W}_{Q\,(:,i)}^{(1)}}^{d\times 1}\overbrace{\boldsymbol{W}_{K\,(:,i)}^{(1)}{}^{\top}}^{1\times d}\overbrace{\boldsymbol{X}^{(0)}{}^{\top}}^{d\times S}/\sqrt{d})\Big)^{\top}}_{S\times 1}\underbrace{\mathrm{smx}(\overbrace{\boldsymbol{X}_{(j,:)}^{(0)}}^{1\times d}\overbrace{\boldsymbol{W}_{Q\,(:,i)}^{(1)}}^{d\times 1}\overbrace{\boldsymbol{W}_{K\,(:,i)}^{(1)}{}^{\top}}^{1\times d}\overbrace{\boldsymbol{X}^{(0)}{}^{\top}}^{d\times S}/\sqrt{d})}_{1\times S}\Bigg)
$$

$$
= \underbrace{\boldsymbol{X}_{(j,:)}^{(0)}}_{1\times d} \cdot \Bigg( \underbrace{d^{-1/2}}_{1\times 1}\underbrace{\mathcal{K}_{(:,i)}^{(1)}{}^{\top}}_{1\times S}\mathrm{blkdiag}\Big(\mathrm{smx}(\overbrace{\mathcal{Q}_{(j,i)}^{(1)}}^{1\times 1}\overbrace{\mathcal{K}_{(:,i)}^{(1)}{}^{\top}}^{1\times S}/\sqrt{d})\Big)
$$
$$
\underbrace{\phantom{\mathrm{blkdiag}\Big(\mathrm{smx}(\mathcal{Q}^{(1)})\Big)}}_{S\times S}
$$
$$
-\underbrace{d^{-1/2}}_{1\times 1}\underbrace{\mathcal{K}_{(:,i)}^{(1)}{}^{\top}}_{1\times S}\underbrace{\Big(\mathrm{smx}(\overbrace{\mathcal{Q}_{(j,i)}^{(1)}}^{1\times 1}\overbrace{\mathcal{K}_{(:,i)}^{(1)}{}^{\top}}^{1\times S}/\sqrt{d})\Big)^{\top}}_{S\times 1}\underbrace{\mathrm{smx}(\overbrace{\mathcal{Q}_{(j,i)}^{(1)}}^{1\times 1}\overbrace{\mathcal{K}_{(:,i)}^{(1)}{}^{\top}}^{1\times S}/\sqrt{d})}_{1\times S}\Bigg)
$$

$$
= d^{-1/2}\underbrace{\boldsymbol{X}_{(j,:)}^{(0)}}_{1\times d} \cdot \underbrace{\Bigg(\overbrace{\mathcal{K}_{(:,i)}^{(1)}{}^{\top}}^{1\times S}\overbrace{\mathrm{blkdiag}\Big(\mathrm{smx}(\xi_{(j,i;:,i)}^{(1)})\Big)}^{S\times S} - \overbrace{\mathcal{K}_{(:,i)}^{(1)}{}^{\top}}^{1\times S}\overbrace{\big(\mathrm{smx}(\xi_{(j,i;:,i)}^{(1)})\big)^{\top}\mathrm{smx}(\xi_{(j,i;:,i)}^{(1)})}^{S\times S}\Bigg)}_{1\times S},
$$

$$(24)$$

where $\mathcal{Q}^{(\ell)} \coloneqq \boldsymbol{X}^{(\ell-1)}\boldsymbol{W}_Q^{(\ell)}$, $\mathcal{K}^{(\ell)} \coloneqq \boldsymbol{X}^{(\ell-1)}\boldsymbol{W}_K^{(\ell)}$, and $\xi_{(j,i;:,i)}^{(1)} \coloneqq \mathcal{Q}_{(j,i)}^{(1)}\mathcal{K}_{(:,i)}^{(1)}{}^{\top}/\sqrt{d}$.

Integrating Equations 22, 23, and 24, and dropping the superscripts for $\mathcal{Q}$ and $\mathcal{K}$ for brevity, yields:

$$
\frac{\partial \boldsymbol{X}_{\mathrm{TF}}^{(2)}}{\partial \boldsymbol{W}_{Q\,(:,i)}^{(1)}}
$$

$$
= \begin{bmatrix} \dot{\sigma}_1/\sqrt{d}\,\underbrace{\boldsymbol{X}_{(1,:)}^{(0)}}_{1\times d} \cdot \underbrace{\Big(\overbrace{\mathcal{K}_{(:,i)}^{(1)}{}^{\top}}^{1\times S}\overbrace{\mathrm{blkdiag}\big(\mathrm{smx}(\xi_{(1,i;:,i)}^{(1)})\big)}^{S\times S}\overbrace{\boldsymbol{X}^{(0)}}^{S\times d}\overbrace{\boldsymbol{W}_V^{(1,1)}}^{d\times h_1}\overbrace{\boldsymbol{W}^{(2)}}^{h_1\times 1} - \overbrace{\mathcal{K}_{(:,i)}^{(1)}{}^{\top}}^{1\times S}\overbrace{\big(\mathrm{smx}(\xi_{(1,i;:,i)}^{(1)})\big)^{\top}\mathrm{smx}(\xi_{(1,i;:,i)}^{(1)})}^{S\times S}\overbrace{\boldsymbol{X}^{(0)}}^{S\times d}\overbrace{\boldsymbol{W}_V^{(1,1)}}^{d\times h_1}\overbrace{\boldsymbol{W}^{(2)}}^{h_1\times 1}\Big)}_{1\times 1} \\ \cdots \\ \dot{\sigma}_S/\sqrt{d}\,\underbrace{\boldsymbol{X}_{(S,:)}^{(0)}}_{1\times d} \cdot \underbrace{\Big(\overbrace{\mathcal{K}_{(:,i)}^{(1)}{}^{\top}}^{1\times S}\overbrace{\mathrm{blkdiag}\big(\mathrm{smx}(\xi_{(S,i;:,i)}^{(1)})\big)}^{S\times S}\overbrace{\boldsymbol{X}^{(0)}}^{S\times d}\overbrace{\boldsymbol{W}_V^{(1,1)}}^{d\times h_1}\overbrace{\boldsymbol{W}^{(2)}}^{h_1\times 1} - \overbrace{\mathcal{K}_{(:,i)}^{(1)}{}^{\top}}^{1\times S}\overbrace{\big(\mathrm{smx}(\xi_{(S,i;:,i)}^{(1)})\big)^{\top}\mathrm{smx}(\xi_{(S,i;:,i)}^{(1)})}^{S\times S}\overbrace{\boldsymbol{X}^{(0)}}^{S\times d}\overbrace{\boldsymbol{W}_V^{(1,1)}}^{d\times h_1}\overbrace{\boldsymbol{W}^{(2)}}^{h_1\times 1}\Big)}_{1\times 1} \end{bmatrix}_{S\times d}
$$

$$
= \Big[ \dot{\sigma}_j/\sqrt{d}\,\boldsymbol{X}_{(j,:)}^{(0)} \cdot \Big( \mathcal{K}_{(:,i)}{}^{\top}\mathrm{blkdiag}\big(\mathrm{smx}(\xi_{(j,i;:,i)}^{(1)})\big) - \mathcal{K}_{(:,i)}{}^{\top}\big(\mathrm{smx}(\xi_{(j,i;:,i)}^{(1)})\big)^{\top}\mathrm{smx}(\xi_{(j,i;:,i)}^{(1)})\Big) \cdot \boldsymbol{X}^{(0)}\boldsymbol{W}_V^{(1,1)}\boldsymbol{W}^{(2)} \Big]_{S\times d}
$$

$$
= \mathrm{blkdiag}\Big( \dot{\sigma}_1/\sqrt{d}\,\Big( \mathcal{K}_{(:,i)}{}^{\top}\mathrm{blkdiag}\big(\mathrm{smx}(\xi_{(1,i;:,i)}^{(1)})\big) - \mathcal{K}_{(:,i)}{}^{\top}\big(\mathrm{smx}(\xi_{(1,i;:,i)}^{(1)})\big)^{\top}\mathrm{smx}(\xi_{(1,i;:,i)}^{(1)})\Big)\boldsymbol{X}^{(0)}\boldsymbol{W}_V^{(1,1)}\boldsymbol{W}^{(2)},
$$

$$
\cdots, \dot{\sigma}_S/\sqrt{d}\,\Big( \mathcal{K}_{(:,i)}{}^{\top}\mathrm{blkdiag}\big(\mathrm{smx}(\xi_{(S,i;:,i)}^{(1)})\big) - \mathcal{K}_{(:,i)}{}^{\top}\big(\mathrm{smx}(\xi_{(S,i;:,i)}^{(1)})\big)^{\top}\mathrm{smx}(\xi_{(S,i;:,i)}^{(1)})\Big)\boldsymbol{X}^{(0)}\boldsymbol{W}_V^{(1,1)}\boldsymbol{W}^{(2)} \Big)\boldsymbol{X}^{(0)}.
$$

$$(25)$$

The derivative with respect to the **key weight matrix** follows a parallel approach. For $i \in \mathbb{N}_p$,

$$
\frac{\partial \boldsymbol{X}_{\text{TF}}^{(2)}}{\partial \boldsymbol{W}_{K(:,i)}^{(1)}} = \frac{\partial \sigma\left(\text{smx}\left(d^{-1/2}\boldsymbol{X}^{(0)}\boldsymbol{W}_Q^{(1)}\left(\boldsymbol{X}^{(0)}\boldsymbol{W}_K^{(1)}\right)^\top\right)\boldsymbol{X}^{(0)}\boldsymbol{W}_V^{(1,1)}\right) \cdot \boldsymbol{W}^{(2)}}{\partial \boldsymbol{W}_{K(:,i)}^{(1)}}
$$

$$
= \dot{\sigma}\frac{\partial \text{smx}\left(d^{-1/2}\boldsymbol{X}^{(0)}\boldsymbol{W}_Q^{(1)}\left(\boldsymbol{X}^{(0)}\boldsymbol{W}_K^{(1)}\right)^\top\right)\boldsymbol{X}^{(0)}\boldsymbol{W}_V^{(1,1)}}{\partial \boldsymbol{W}_{K(:,i)}^{(1)}} \cdot \boldsymbol{W}^{(2)}
$$

$$
= \dot{\sigma}\frac{\partial \text{smx}\left(d^{-1/2}\boldsymbol{X}^{(0)}\boldsymbol{W}_Q^{(1)}\left(\boldsymbol{X}^{(0)}\boldsymbol{W}_K^{(1)}\boldsymbol{e}_i\boldsymbol{e}_i^\top\right)^\top\right)\boldsymbol{X}^{(0)}\boldsymbol{W}_V^{(1,1)}}{\partial \boldsymbol{W}_{K(:,i)}^{(1)}} \cdot \boldsymbol{W}^{(2)}
$$

$$
= \begin{bmatrix} \dot{\sigma}_1 \dfrac{\partial \text{smx}\left(d^{-1/2}\boldsymbol{X}_{(1,:)}^{(0)}\boldsymbol{W}_{Q(:,i)}^{(1)}\boldsymbol{W}_{K(:,i)}^{(1)}{}^\top\boldsymbol{X}^{(0)\top}\right)\boldsymbol{X}^{(0)}\boldsymbol{W}_V^{(1,1)}}{\partial \boldsymbol{W}_{K(:,i)}^{(1)}} \cdot \boldsymbol{W}^{(2)} \\ \cdots \\ \dot{\sigma}_S \dfrac{\partial \text{smx}\left(d^{-1/2}\boldsymbol{X}_{(S,:)}^{(0)}\boldsymbol{W}_{Q(:,i)}^{(1)}\boldsymbol{W}_{K(:,i)}^{(1)}{}^\top\boldsymbol{X}^{(0)\top}\right)\boldsymbol{X}^{(0)}\boldsymbol{W}_V^{(1,1)}}{\partial \boldsymbol{W}_{K(:,i)}^{(1)}} \cdot \boldsymbol{W}^{(2)} \end{bmatrix}
$$

$$
= \begin{bmatrix} \dot{\sigma}_1 \dfrac{\partial \text{smx}\left(d^{-1/2}\boldsymbol{X}_{(1,:)}^{(0)}\boldsymbol{W}_{Q(:,i)}^{(1)}\boldsymbol{W}_{K(:,i)}^{(1)}{}^\top\boldsymbol{X}^{(0)\top}\right)}{\partial \boldsymbol{W}_{K(:,i)}^{(1)}} \cdot \boldsymbol{X}^{(0)}\boldsymbol{W}_V^{(1,1)}\boldsymbol{W}^{(2)} \\ \cdots \\ \dot{\sigma}_S \dfrac{\partial \text{smx}\left(d^{-1/2}\boldsymbol{X}_{(S,:)}^{(0)}\boldsymbol{W}_{Q(:,i)}^{(1)}\boldsymbol{W}_{K(:,i)}^{(1)}{}^\top\boldsymbol{X}^{(0)\top}\right)}{\partial \boldsymbol{W}_{K(:,i)}^{(1)}} \cdot \boldsymbol{X}^{(0)}\boldsymbol{W}_V^{(1,1)}\boldsymbol{W}^{(2)} \end{bmatrix} \tag{26}
$$

Again, examining row by row for the $j$-th row ($j \in \mathbb{N}_S$) in Equation 26, the focal derivative is $\frac{\partial \text{smx}\left(d^{-1/2}\boldsymbol{X}_{(j,:)}^{(0)}\boldsymbol{W}_{Q(:,i)}^{(1)}\boldsymbol{W}_{K(:,i)}^{(1)}{}^\top\boldsymbol{X}^{(0)\top}\right)}{\partial \boldsymbol{W}_{K(:,i)}^{(1)}}$:

$$
\frac{\partial \text{smx}\left(d^{-1/2}\boldsymbol{X}_{(j,:)}^{(0)}\boldsymbol{W}_{Q(:,i)}^{(1)}\boldsymbol{W}_{K(:,i)}^{(1)}{}^\top\boldsymbol{X}^{(0)\top}\right)}{\partial \boldsymbol{W}_{K(:,i)}^{(1)}}
$$

$$
= \frac{\partial d^{-1/2}\boldsymbol{X}_{(j,:)}^{(0)}\boldsymbol{W}_{Q(:,i)}^{(1)}\boldsymbol{W}_{K(:,i)}^{(1)}{}^\top\boldsymbol{X}^{(0)\top}}{\partial \boldsymbol{W}_{K(:,i)}^{(1)}}\frac{\partial \text{smx}\left(d^{-1/2}\boldsymbol{X}_{(j,:)}^{(0)}\boldsymbol{W}_{Q(:,i)}^{(1)}\boldsymbol{W}_{K(:,i)}^{(1)}{}^\top\boldsymbol{X}^{(0)\top}\right)}{\partial d^{-1/2}\boldsymbol{X}_{(j,:)}^{(0)}\boldsymbol{W}_{Q(:,i)}^{(1)}\boldsymbol{W}_{K(:,i)}^{(1)}{}^\top\boldsymbol{X}^{(0)\top}}
$$

$$
= \boldsymbol{X}_{(j,:)}^{(0)} \cdot \frac{1}{\sqrt{d}}\boldsymbol{W}_{Q(:,i)}^{(1)}{}^\top\boldsymbol{X}^{(0)\top}\frac{\partial \text{smx}\left(d^{-1/2}\boldsymbol{X}_{(j,:)}^{(0)}\boldsymbol{W}_{Q(:,i)}^{(1)}\boldsymbol{W}_{K(:,i)}^{(1)}{}^\top\boldsymbol{X}^{(0)\top}\right)}{\partial d^{-1/2}\boldsymbol{X}_{(j,:)}^{(0)}\boldsymbol{W}_{Q(:,i)}^{(1)}\boldsymbol{W}_{K(:,i)}^{(1)}{}^\top\boldsymbol{X}^{(0)\top}}
$$

$$
= \boldsymbol{X}_{(j,:)}^{(0)} \cdot \frac{1}{\sqrt{d}}\boldsymbol{W}_{Q(:,i)}^{(1)}{}^\top\boldsymbol{X}^{(0)\top}\frac{\partial \left(\dfrac{\exp\left(d^{-1/2}\boldsymbol{X}_{(j,:)}^{(0)}\boldsymbol{W}_{Q(:,i)}^{(1)}\boldsymbol{W}_{K(:,i)}^{(1)}{}^\top\boldsymbol{X}^{(0)\top}\right)}{\mathbf{1}^\top\exp\left(d^{-1/2}\boldsymbol{X}_{(j,:)}^{(0)}\boldsymbol{W}_{Q(:,i)}^{(1)}\boldsymbol{W}_{K(:,i)}^{(1)}{}^\top\boldsymbol{X}^{(0)\top}\right)}\right)}{\partial d^{-1/2}\boldsymbol{X}_{(j,:)}^{(0)}\boldsymbol{W}_{Q(:,i)}^{(1)}\boldsymbol{W}_{K(:,i)}^{(1)}{}^\top\boldsymbol{X}^{(0)\top}}
$$

$$
= \boldsymbol{X}_{(j,:)}^{(0)} \cdot \left(\frac{1}{\sqrt{d}}\boldsymbol{W}_{Q(:,i)}^{(1)}{}^\top\boldsymbol{X}^{(0)\top}\text{blkdiag}\left(\text{smx}(\boldsymbol{X}_{(j,:)}^{(0)}\boldsymbol{W}_{Q(:,i)}^{(1)}\boldsymbol{W}_{K(:,i)}^{(1)}{}^\top\boldsymbol{X}^{(0)\top}/\sqrt{d})\right)\right.
$$
$$
\left. -\frac{1}{\sqrt{d}}\boldsymbol{W}_{Q(:,i)}^{(1)}{}^\top\boldsymbol{X}^{(0)\top}\left(\text{smx}(\boldsymbol{X}_{(j,:)}^{(0)}\boldsymbol{W}_{Q(:,i)}^{(1)}\boldsymbol{W}_{K(:,i)}^{(1)}{}^\top\boldsymbol{X}^{(0)\top}/\sqrt{d})\right)^\top\text{smx}(\boldsymbol{X}_{(j,:)}^{(0)}\boldsymbol{W}_{Q(:,i)}^{(1)}\boldsymbol{W}_{K(:,i)}^{(1)}{}^\top\boldsymbol{X}^{(0)\top}/\sqrt{d})\right)
$$
$$\tag{27}$$

Simplifying the right-hand side of Equation 27 gives:

$$
\boldsymbol{X}_{(j,:)}^{(0)} \cdot \left( \underbrace{d^{-1/2}}_{1\times 1} \underbrace{\boldsymbol{W}_{Q\,(:,i)}^{(1)}}_{1\times d}{}^{\top} \overbrace{\boldsymbol{X}^{(0)}{}^{\top}}^{d\times S} \mathrm{blkdiag}\Big( \mathrm{smx}(\overbrace{\boldsymbol{X}_{(j,:)}^{(0)}}^{1\times d} \overbrace{\boldsymbol{W}_{Q\,(:,i)}^{(1)}}^{d\times 1} \overbrace{\boldsymbol{W}_{K\,(:,i)}^{(1)}{}^{\top}}^{1\times d} \overbrace{\boldsymbol{X}^{(0)}{}^{\top}}^{d\times S} /\sqrt{d}) \Big) }_{S\times S}
$$
$$
\underset{\text{size: } 1\times d}{}
$$

$$
- \underbrace{d^{-1/2}}_{1\times 1} \underbrace{\boldsymbol{W}_{Q\,(:,i)}^{(1)}}_{1\times d}{}^{\top} \overbrace{\boldsymbol{X}^{(0)}{}^{\top}}^{d\times S} \underbrace{\Big( \mathrm{smx}(\boldsymbol{X}_{(j,:)}^{(0)} \boldsymbol{W}_{Q\,(:,i)}^{(1)} \boldsymbol{W}_{K\,(:,i)}^{(1)}{}^{\top} \boldsymbol{X}^{(0)}{}^{\top}/\sqrt{d}) \Big)^{\top}}_{S\times 1} \underbrace{\mathrm{smx}(\boldsymbol{X}_{(j,:)}^{(0)} \boldsymbol{W}_{Q\,(:,i)}^{(1)} \boldsymbol{W}_{K\,(:,i)}^{(1)}{}^{\top} \boldsymbol{X}^{(0)}{}^{\top}/\sqrt{d})}_{1\times S} \Big)
$$

$$
= \; \underbrace{\boldsymbol{X}_{(j,:)}^{(0)}}_{1\times d} \cdot \left( \underbrace{d^{-1/2}}_{1\times 1} \underbrace{\mathcal{Q}_{(:,i)}^{(1)}{}^{\top}}_{1\times S} \underbrace{\mathrm{blkdiag}\Big( \mathrm{smx}(\overbrace{\mathcal{Q}_{(j,i)}^{(1)}}^{1\times 1} \overbrace{\mathcal{K}_{(:,i)}^{(1)}{}^{\top}}^{1\times S} /\sqrt{d}) \Big)}_{S\times S} \right.
$$

$$
\left. - \underbrace{d^{-1/2}}_{1\times 1} \underbrace{\mathcal{Q}_{(:,i)}^{(1)}{}^{\top}}_{1\times S} \underbrace{\Big( \mathrm{smx}(\overbrace{\mathcal{Q}_{(j,i)}^{(1)}}^{1\times 1} \overbrace{\mathcal{K}_{(:,i)}^{(1)}{}^{\top}}^{1\times S} /\sqrt{d}) \Big)^{\top}}_{S\times 1} \underbrace{\mathrm{smx}(\overbrace{\mathcal{Q}_{(j,i)}^{(1)}}^{1\times 1} \overbrace{\mathcal{K}_{(:,i)}^{(1)}{}^{\top}}^{1\times S} /\sqrt{d})}_{1\times S} \right)
$$

$$
= \; d^{-1/2} \underbrace{\boldsymbol{X}_{(j,:)}^{(0)}}_{1\times d} \cdot \underbrace{\left( \overbrace{\mathcal{Q}_{(:,i)}^{(1)}{}^{\top}}^{1\times S} \overbrace{\mathrm{blkdiag}\big( \mathrm{smx}(\xi_{(j,i;:,i)}^{(1)}) \big)}^{S\times S} - \overbrace{\mathcal{Q}_{(:,i)}^{(1)}{}^{\top}}^{1\times S} \overbrace{\big( \mathrm{smx}(\xi_{(j,i;:,i)}^{(1)}) \big)^{\top} \mathrm{smx}(\xi_{(j,i;:,i)}^{(1)})}^{S\times S} \right)}_{1\times S},
$$

$$(28)$$

Combining Equations 26, 27, and 28 results in:

$$
\frac{\partial \boldsymbol{X}_{\mathrm{TF}}^{(2)}}{\partial \boldsymbol{W}_{K\,(:,i)}^{(1)}}
$$

$$
= \begin{bmatrix} \dot{\sigma}_1/\sqrt{d}\, \underbrace{\boldsymbol{X}_{(1,:)}^{(0)}}_{1\times d} \cdot \underbrace{\Big( \overbrace{\mathcal{Q}_{(:,i)}^{(1)}{}^{\top}}^{1\times S} \overbrace{\mathrm{blkdiag}\big( \mathrm{smx}(\xi_{(1,i;:,i)}^{(1)}) \big)}^{S\times S} \overbrace{\boldsymbol{X}^{(0)}}^{S\times d} \overbrace{\boldsymbol{W}_V^{(1,1)}}^{d\times h_1} \overbrace{\boldsymbol{W}^{(2)}}^{h_1\times 1} - \overbrace{\mathcal{Q}_{(:,i)}^{(1)}{}^{\top}}^{1\times S} \overbrace{\big( \mathrm{smx}(\xi_{(1,i;:,i)}^{(1)}) \big)^{\top} \mathrm{smx}(\xi_{(1,i;:,i)}^{(1)})}^{S\times S} \overbrace{\boldsymbol{X}^{(0)}}^{S\times d} \overbrace{\boldsymbol{W}_V^{(1,1)}}^{d\times h_1} \overbrace{\boldsymbol{W}^{(2)}}^{h_1\times 1} \Big)}_{1\times 1} \\ \cdots \\ \dot{\sigma}_S/\sqrt{d}\, \underbrace{\boldsymbol{X}_{(S,:)}^{(0)}}_{1\times d} \cdot \underbrace{\Big( \overbrace{\mathcal{Q}_{(:,i)}^{(1)}{}^{\top}}^{1\times S} \overbrace{\mathrm{blkdiag}\big( \mathrm{smx}(\xi_{(S,i;:,i)}^{(1)}) \big)}^{S\times S} \overbrace{\boldsymbol{X}^{(0)}}^{S\times d} \overbrace{\boldsymbol{W}_V^{(1,1)}}^{d\times h_1} \overbrace{\boldsymbol{W}^{(2)}}^{h_1\times 1} - \overbrace{\mathcal{Q}_{(:,i)}^{(1)}{}^{\top}}^{1\times S} \overbrace{\big( \mathrm{smx}(\xi_{(S,i;:,i)}^{(1)}) \big)^{\top} \mathrm{smx}(\xi_{(S,i;:,i)}^{(1)})}^{S\times S} \overbrace{\boldsymbol{X}^{(0)}}^{S\times d} \overbrace{\boldsymbol{W}_V^{(1,1)}}^{d\times h_1} \overbrace{\boldsymbol{W}^{(2)}}^{h_1\times 1} \Big)}_{1\times 1} \end{bmatrix}_{S\times d}
$$

$$
= \; \Big[ \dot{\sigma}_j/\sqrt{d}\, \boldsymbol{X}_{(j,:)}^{(0)} \cdot \Big( \mathcal{Q}_{(:,i)}^{(1)}{}^{\top} \mathrm{blkdiag}\big( \mathrm{smx}(\xi_{(j,i;:,i)}^{(1)}) \big) - \mathcal{Q}_{(:,i)}^{(1)}{}^{\top} \big( \mathrm{smx}(\xi_{(j,i;:,i)}^{(1)}) \big)^{\top} \mathrm{smx}(\xi_{(j,i;:,i)}^{(1)}) \Big) \cdot \boldsymbol{X}^{(0)} \boldsymbol{W}_V^{(1,1)} \boldsymbol{W}^{(2)} \Big]_{S\times d}
$$

$$
= \; \mathrm{blkdiag}\Big( \dot{\sigma}_1/\sqrt{d}\, \Big( \mathcal{Q}_{(:,i)}^{(1)}{}^{\top} \mathrm{blkdiag}\big( \mathrm{smx}(\xi_{(1,i;:,i)}^{(1)}) \big) - \mathcal{Q}_{(:,i)}^{(1)}{}^{\top} \big( \mathrm{smx}(\xi_{(1,i;:,i)}^{(1)}) \big)^{\top} \mathrm{smx}(\xi_{(1,i;:,i)}^{(1)}) \Big) \boldsymbol{X}^{(0)} \boldsymbol{W}_V^{(1,1)} \boldsymbol{W}^{(2)},
$$

$$
\cdots, \dot{\sigma}_S/\sqrt{d}\, \Big( \mathcal{Q}_{(:,i)}^{(1)}{}^{\top} \mathrm{blkdiag}\big( \mathrm{smx}(\xi_{(S,i;:,i)}^{(1)}) \big) - \mathcal{Q}_{(:,i)}^{(1)}{}^{\top} \big( \mathrm{smx}(\xi_{(S,i;:,i)}^{(1)}) \big)^{\top} \mathrm{smx}(\xi_{(S,i;:,i)}^{(1)}) \Big) \boldsymbol{X}^{(0)} \boldsymbol{W}_V^{(1,1)} \boldsymbol{W}^{(2)} \Big) \boldsymbol{X}^{(0)}.
$$

$$(29)$$

These derivations naturally extend to scenarios where each component of the output $\boldsymbol{X}_{\mathrm{TF}}^{(L)}$ is a vector, through a multi-dimensional framework (Micchelli & Pontil, 2005; Zhang et al., 2023). For multi-head attention, the process can be parallelized by replicating the derivations across the number of heads.

## A.5 FIGHTER FORWARD PASS (WITH OPTIONAL ENHANCEMENTS)

---

**Algorithm 1** Fighter Forward Pass (with Optional Enhancements)

---

**Require:** Input features $\boldsymbol{X}_{\text{FT}}^{(0)} \in \mathbb{R}^{S \times d}$, temporary variable $\boldsymbol{A}$, number of layers $L$, hop parameters $\{\kappa_\ell\}_{\ell=1}^L$, weights $\{\boldsymbol{W}_Q^{(\ell)}, \boldsymbol{W}_K^{(\ell)}, \boldsymbol{W}^{(\ell)}\}_{\ell=1}^L$

**Ensure:** Output $\boldsymbol{X}_{\text{FT}}^{(L)}$

1: $\boldsymbol{X} \leftarrow \boldsymbol{X}_{\text{FT}}^{(0)}$              ▷ Initialize with input features
2: **for** $\ell = 1$ to $L - 1$ **do**
3:   *(Optional)*: $\tilde{\boldsymbol{X}} \leftarrow \text{LayerNorm}(\boldsymbol{X})$       ▷ Apply layer normalization
4:   $\boldsymbol{Q} \leftarrow \tilde{\boldsymbol{X}} \boldsymbol{W}_Q^{(\ell)}$          ▷ Queries; use $\boldsymbol{X}$ if no norm
5:   $\boldsymbol{K} \leftarrow \tilde{\boldsymbol{X}} \boldsymbol{W}_K^{(\ell)}$          ▷ Keys; use $\boldsymbol{X}$ if no norm
6:   *(Optional)*: Compute $\boldsymbol{Q}, \boldsymbol{K}$ for each head, concatenate/project outputs
7:   $\boldsymbol{A} \leftarrow \text{smx}(d^{-1/2} \boldsymbol{Q} \boldsymbol{K}^\top)$     ▷ Compute attention distribution matrices
8:   $\tilde{\boldsymbol{A}} \leftarrow \boldsymbol{A}^{[\kappa_\ell]}$       ▷ Multi-hop attention: raise to powers up to $\kappa_\ell - 1$
9:   $\tilde{\boldsymbol{X}} \leftarrow \text{blkdiag}(\tilde{\boldsymbol{X}}; \kappa_\ell)$     ▷ Block-diagonal replication; use $\boldsymbol{X}$ if no norm
10:   $\tilde{\boldsymbol{X}} \leftarrow \tilde{\boldsymbol{A}} \tilde{\boldsymbol{X}} \boldsymbol{W}^{(\ell)}$     ▷ Aggregate and project with streamlined weight
11:   $\boldsymbol{X} \leftarrow \sigma(\tilde{\boldsymbol{X}})$          ▷ Apply activation (*e.g.*, ReLU)
12:   *(Optional)*: $\boldsymbol{X} \leftarrow \boldsymbol{X} + \tilde{\boldsymbol{X}}$      ▷ Add residual; use $\boldsymbol{X}$ if no norm
13: **end for**
14: *(Optional)*: $\tilde{\boldsymbol{X}} \leftarrow \text{LayerNorm}(\boldsymbol{X})$       ▷ Final layer normalization
15: $\boldsymbol{Q} \leftarrow \tilde{\boldsymbol{X}} \boldsymbol{W}_Q^{(L)}$         ▷ Final queries; use $\boldsymbol{X}$ if no norm
16: $\boldsymbol{K} \leftarrow \tilde{\boldsymbol{X}} \boldsymbol{W}_K^{(L)}$          ▷ Final keys; use $\boldsymbol{X}$ if no norm
17: *(Optional)*: Compute $\boldsymbol{Q}, \boldsymbol{K}$ for each head, concatenate/project outputs
18: $\boldsymbol{A} \leftarrow \text{smx}(d^{-1/2} \boldsymbol{Q} \boldsymbol{K}^\top)$      ▷ Final attention distribution matrices
19: $\tilde{\boldsymbol{A}} \leftarrow \boldsymbol{A}^{[\kappa_\ell]}$         ▷ Final multi-hop attention
20: $\tilde{\boldsymbol{X}} \leftarrow \text{blkdiag}(\tilde{\boldsymbol{X}}; \kappa_\ell)$     ▷ Final block-diagonal replication; use $\boldsymbol{X}$ if no norm
21: $\boldsymbol{X}_{\text{FT}}^{(L)} \leftarrow \tilde{\boldsymbol{A}}^{[\kappa_L]} \tilde{\boldsymbol{X}} \boldsymbol{W}^{(L)}$         ▷ Final output
22: *(Optional)*: $\boldsymbol{X}_{\text{FT}}^{(L)} \leftarrow \boldsymbol{X}_{\text{FT}}^{(L)} + \tilde{\boldsymbol{X}}$    ▷ Add final residual; use $\boldsymbol{X}$ if no norm **return** $\boldsymbol{X}_{\text{FT}}^{(L)}$

---

## B EXPERIMENT DETAILS

**Device Setup.** We mainly conduct experiments using NVIDIA Geforce RTX 3090 (24G).

**Dataset Splitting.** The used datasets are:

- Electricity[3]: containing the hourly electricity consumption of 321 customers from 2012 to 2014;

- Weather[4]: containing 21 meteorological indicators, recorded every 10 minutes for the 2020 year;

- ETT(Zhou et al., 2021): containing 7 electricity transformer factors, ETTh1 is recorded every hour.

The train / val / test split configurations for the benchmark datasets are provided in Table 3. All time series inputs are standardized using a StandardScaler(Pedregosa et al., 2011).

| Dataset | train | validation | test |
|---|---|---|---|
| Electricity | 70% | 10% | 20% |
| Weather | 70% | 10% | 20% |
| ETTh1 | 60% | 20% | 20% |

Table 3: Dataset splitting for the benchmark datasets.

**Hyperparameter Settings.** The key hyperparameter settings are listed in Table 4. The prediction lengths for Electricity and Weather are given {96, 192, 336}, and the prediction lengths for ETTh1 are given 720.

| Dataset | n-blocks | $\kappa-$list | batch-size | input-len | epochs |
|---|---|---|---|---|---|
| Electricity | 2 | [3] | 128 | 96 | 25 |
| Weather | 2 | [3] | 128 | 96 | 25 |
| ETTh1 | 2 | [3] | 128 | 96 | 25 |

Table 4: Key hyperparameter settings on the benchmark datasets for Fighter.

We also adopt the `ReduceLROnPlateau` (Hinton et al., 2012) as the learning rate scheduler.

---

[3] https://archive.ics.uci.edu/ml/datasets/ElectricityLoadDiagrams20112014
[4] https://www.bgc-jena.mpg.de/wetter

