# OpenReview forum: "Fighter: Unveiling the Graph Convolutional Nature of Transformers in Time Series Modeling"
_ICLR.cc/2026/Workshop/Sci4DL — Submitted to Sci4DL 2026_

### Official Review · Reviewer_3req · 2026-02-26

**Fit:** 2
**Significance:** 2
**Confidence:** 2

**Summary:**

The paper investigates the theoretical equivalence between Transformer encoders and Graph Convolutional Networks for time series forecasting The authors mathematically demonstrate that the self attention distribution matrix functions as a dynamic and learnable adjacency matrix Building on this connection they propose Fighter which is a modified architecture that eliminates redundant feed forward linear projections and integrates explicit multi hop graph aggregation to model temporal dependencies across multiple scales

**Strengths:**

* The formal mapping of both the forward and backward passes of Transformers to GCNs offers a clean mechanistic interpretation of how attention learns temporal dependencies
* Removing the secondary projection based on the GCN equivalence is an elegant architectural pruning step
* While the high level analogy between Transformers and GNNs is known in the broader literature the specific application of dynamic multi hop aggregation to the temporal axis is an interesting scope to consider and presented well by authors

**Suggestions:**

* The broader premise that Transformers are essentially GNNs operating on fully connected graphs has been discussed in the community for several years which limits the foundational novelty as briefly touched upon in lines 164 to 165 regarding existing studies treating sequences as graphs
Weakness:
* The mathematical exposition is dense and lacks intuition for non theoretical readers particularly around the backward pass gradient derivations spanning lines 430 to 519 where the complex equations for feature transformation parameters and query key weight matrices are introduced
* The original manuscript lacked a dedicated computational complexity analysis in the main text experiments section spanning lines 631 to 817 which makes it difficult to assess the practical trade offs of the multi hop mechanism without diving into the newly added appendix starting at line 2165
* The empirical section required heavy patching during the ICLR main conference rebuttal phase to fix train validation test splits which makes the current polished version presented in Table 1 feel slightly fragile

Suggestion/Question:
I would be interested to hear how sensitive the multi hop aggregation parameter is to varying sampling rates or missing data in noisy real world datasets It would also be great to know how Fighter practically scales on extremely long context windows compared to state of the art sub quadratic attention mechanisms.

---

### Meta-Review · Area_Chair_j5Df · 2026-03-02

**Recommendation:** Reject

**Metareview:**

This work focuses on theoretical equivalence between Transformer encoders and GCNs for time series forecasting. The paper is the current form does not seem very accessible to the community. It would be helpful if the paper presentation could be improved to highlight the work done as well as fully distinguish from previous work that have similar focus.
Due to this, I recommend a rejection.

---

### Decision · Program_Chairs · 2026-03-02

Reject